# Structures of human prostaglandin $F_{2\alpha}$ receptor reveal the mechanism of ligand and G protein selectivity

Xiuqing Lv[1,8], Kaixuan Gao[2,3,8], Jia Nie[1,8], Xin Zhang[2,3], Shuhao Zhang[2,3], Yinhang Ren[2,3], Xiaoou Sun[3,4], Qi Li[5], Jingrui Huang [1], Lijuan Liu[1], Xiaowen Zhang[1], Weishe Zhang [1,6] ✉ & Xiangyu Liu [2,3,7] ✉

Prostaglandins and their receptors regulate various physiological processes. Carboprost, an analog of prostaglandin $F_{2\alpha}$ and an agonist for the prostaglandin F2-alpha receptor (FP receptor), is clinically used to treat postpartum hemorrhage (PPH). However, off-target activation of closely related receptors such as the prostaglandin E receptor subtype EP3 (EP3 receptor) by carboprost results in side effects and limits the clinical application. Meanwhile, the FP receptor selective agonist latanoprost is not suitable to treat PPH due to its poor solubility and fast clearance. Here, we present two cryo-EM structures of the FP receptor bound to carboprost and latanoprost-FA (the free acid form of latanoprost) at 2.7 Å and 3.2 Å resolution, respectively. The structures reveal the molecular mechanism of FP receptor selectivity for both endogenous prostaglandins and clinical drugs, as well as the molecular mechanism of G protein coupling preference by the prostaglandin receptors. The structural information may guide the development of better prostaglandin drugs.

Prostaglandins, the metabolic products of arachidonates, regulate various physiologic functions through interacting and activating nine different prostaglandin receptors that belong to the G protein coupled receptor family. The endogenous prostaglandins include Prostaglandin $F_{2\alpha}$ ($PGF_{2\alpha}$), Prostaglandin $E_2$ ($PGE_2$), Prostaglandin $D_2$ ($PGD_2$), Prostaglandin $I_2$ ($PGI_2$), and thromboxane $A_2$ ($TXA_2$), while the corresponding receptors include prostaglandin $F_{2\alpha}$ (FP) receptor, Prostaglandin $E_2$ (EP1-4) receptors, Prostaglandin $D_2$ (DP1-2) receptors, Prostaglandin $I_2$ (IP) receptor and Thromboxane $A_2$ (TP) receptor. These receptors further activate different subtypes of G protein, with DP1, EP2, EP4, and IP receptors primarily couple to Gs; FP, EP1 and TP receptors primarily couple to Gq and EP3 receptor couples to Gi[1]. The interplay between the prostaglandins, the receptors and downstream G proteins mediate the complex regulation of physiological functions

including cardiovascular homeostasis, body temperature control, female reproduction and inflammation[2-4]. Structural and mechanistic studies on the prostaglandins and prostaglandin receptors have drawn intense attention, the $PGE_2$ bound EP2-Gs complex, $PGE_2$ bound EP3 receptor, $PGE_2$ bound EP4-Gs complex, as well as $PGF_{2\alpha}$ bound FP-Gq complex structures have recently been reported[5-8].

The prostaglandin receptors have different expression profiles and tissue distribution. Among them, the FP receptor is highly expressed in smooth muscle, uterine myometrium and eye[9-11], which makes it a target for the treatment of glaucoma and postpartum hemorrhage (PPH), the severe bleeding after childbirth. PPH poses a major threat to women health and accounts for approximately 18% of all deaths of pregnant women globally[12,13]. The causes of postpartum bleeding include uterine atony, obstetric lacerations, retained

---

[1]Department of Obstetrics, Xiangya Hospital Central South University, Changsha, China. [2]State Key Laboratory of Membrane Biology, Tsinghua-Peking Center for Life Sciences, School of Pharmaceutical Sciences, Tsinghua University, Beijing, China. [3]Beijing Frontier Research Center for Biological Structure, Beijing Advanced Innovation Center for Structural Biology, Tsinghua University, Beijing, China. [4]School of Medicine, Tsinghua University, Beijing, China. [5]Reproductive Medicine Center, Xiangya Hospital Central South University, Changsha, China. [6]Hunan Engineering Research Center of Early Life Development and Disease Prevention, Changsha, China. [7]Beijing Key Laboratory of Cardiovascular Receptors Research, Peking University, Beijing, China. [8]These authors contributed equally: Xiuqing Lv, Kaixuan Gao, Jia Nie. ✉e-mail: zhangweishe@yeah.net; liu_xy@mail.tsinghua.edu.cn

placental tissue, and coagulation disorders. Of all these causes, uterine atony, which is defined as failure of the uterus adequate contraction after placental delivery, contributed to at least 70–80% of PPH[14]. The FP receptor belongs to the class A G protein-coupled receptor (GPCR), and couples to the Gq subtype of G protein[11]. Activation of the FP receptor in the human myometrium by its endogenous agonist prostaglandin F$_{2\alpha}$ (PGF$_{2\alpha}$) causes the elevation of intracellular calcium concentration and leads to contraction of the uterine smooth muscle[15]. Therefore, PGF$_{2\alpha}$ and its analogs have been widely used as therapeutics for uterine atonic PPH in obstetric practice[16,17].

Carboprost tromethamine is a tromethamine salt form of 15-methyl PGF$_{2\alpha}$ and is the most commonly used medicine to treat atonic PPH, especially for severe uterine atonic PPH[17]. While the pharmacological profile of carboprost tromethamine is similar to PGF$_{2\alpha}$, the contraction intensity of the uterus is 20–100 times stronger when treated with carboprost tromethamine than with PGF$_{2\alpha}$[16]. The clinical application of PGF$_{2\alpha}$ is limited by its rapid metabolism. Replacing the hydrogen at carbon 15 with a methyl group in carboprost protects the compound from further oxidation and results in preferred pharmacokinetics properties[18].

Carboprost tromethamine represents the most effective treatment of atonic PPH by activating the FP receptor. However, it has side effects such as fever and hypertension due to its activation of the closely related EP3 receptor[19–21]. The off-target side effects limit its clinical application, especially to patients with cardiovascular diseases. Both carboprost and PGF$_{2\alpha}$ only have 10-fold selectivity towards the FP receptor over the EP3 receptor in NanoBiT assay (Supplementary Fig. 1a, b), which is consistent with previous reports on PGF$_{2\alpha}$'s 10-fold selectivity towards the FP receptor[22]. Improving its selectivity may result in better medications to treat PPH (Fig. 1a).

So far, the most selective agonist for the FP receptor is latanoprost, which is a clinically used drug to treat elevated intraocular pressure in patients with ocular hypertension or open-angle glaucoma. In glaucoma treatment, latanoprost is rapidly absorbed as a prodrug and activated by hydrolysis of the ester bond[23] (Supplementary Fig. 2) to latanoprost free acid (latanoprost-FA). While latanoprost-FA shows 2000-fold selectivity towards the FP receptor over the EP3 receptor[22], its clinical application to treat PPH is limited by its poor solubility and fast clearance rate[24]. Nevertheless, understanding the molecular mechanism of latanoprost-FA's high selectivity towards the FP receptor may guide the optimization of carboprost and other PGF$_{2\alpha}$ analogs. In order to reveal the mechanism of carboprost tromethamine and latanoprost-FA's selectivity towards the FP receptor, we determined the complex structure of the FP receptor bound with these two drugs using cryo-EM method.

## Results

### Overall structure of the FP receptor

To prepare the FP receptor - G protein complexes, we generated a construct with an engineered Gα subunit miniG$_{s/q70iN}$ fused to the C terminus of the FP receptor as previously described[25]. The miniG$_{s/q70iN}$ was constructed by mutating 7 residues on the α5 helix of miniGs to the equivalents from Gαq (R$^{H5.12}$K, Q$^{H5.16}$L, R$^{H5.17}$Q, H$^{H5.19}$N, Q$^{H5.22}$E, E$^{H5.24}$N, and L$^{H5.26}$V), as well as by changing the N-terminus with that of Gαi. The resulted miniG$_{s/q70iN}$ construct recognizes the Gq-coupled receptors[26] and reserves the interaction interface of nanobody35 (Nb35). This miniG$_{s/q70iN}$ fusing strategy improved the protein expression yield and protected the receptor from aggregating during purification. Gβ$_1$γ$_2$ was later added during purification to reconstruct the G protein heterotrimer.

The FP-miniG$_{s/q70iN}$-Gβ$_1$γ$_2$-Nb35 complexes bound to carboprost or latanoprost-FA were purified and their structures were solved using cryo-EM with resolutions of 2.7 Å and 3.2 Å, respectively (Fig. 1b, c, Supplementary Table 3). These two structures are remarkably similar with RMSD of 0.31 Å between Cα atoms. In both structures, the FP

receptor is stabilized at active-state conformation, as revealed by comparison with the active state of the EP3 receptor[6] (PDB 6AK3) and the inactive state of the EP4 receptor[27] (PDB 5YWY) (Supplementary Fig. 3a, b). TM6 exhibits outward displacement compared to the inactive state of the EP4 receptor, the extent of the TM6 displacement is smaller than that of the EP3 receptor (Supplementary Fig. 3a), leading to the different G protein binding orientation, as will be discussed later in more details.

The ECL2 of the FP receptor forms a hairpin structure and covers the orthosteric pocket like a lid, which is also observed in the EP2 receptor, EP3 receptor and EP4 receptor. This is likely a general feature of the prostaglandin receptor family and contributes to the occluded ligand binding pocket of the family[5,6,27] (Supplementary Fig. 4a).

### A conserved prostaglandin-binding pocket in the prostaglandin receptor family

The prostaglandin binding pockets are conserved among the prostaglandin receptor family as revealed by the reported active state structures[5,6,28]. The binding mode of PGE$_2$ to EP3 receptor is most similar to that of carboprost to FP receptor (Supplementary Fig. 4b). In both structures, the ligands adopt an L-shape conformation (Supplementary Fig. 4b). As previously reported[5,6,27], the orthosteric pocket generally contains three sub-pockets covering the α-chain, ω-chain, and F-ring of the ligand (Fig. 2a; Supplementary Fig. 5a). Y92$^{2.65}$, T184$^{ECL2}$ and R291$^{7.40}$ form hydrogen bonds or salt bridges with 1-carboxyl of the α-chain, while M115$^{3.32}$, F187$^{45.51}$, F205$^{5.41}$, W262$^{6.48}$, F265$^{6.51}$ and L290$^{7.39}$ form hydrophobic interactions with the ω-chain (superscripts denote Ballesteros-Weinstein numbering[29], Fig. 2b, c). Additionally, Q297$^{7.46}$ form hydrogen bonds with 15-hydroxyl (Fig. 2c). The above-mentioned residues are conserved in the prostaglandin receptor family except for F205$^{5.41}$, W262$^{6.48}$, F265$^{6.51}$ and Q297$^{7.46}$ (Supplementary Fig. 6). In contrary, the residues interacting with the F-ring are less conserved. S33$^{1.39}$, T294$^{7.43}$ and H81$^{2.54}$ form a sub-pocket to accommodate the 9-hydroxyl and 11-hydroxyl of the F-ring through hydrophilic interactions (Fig. 2d). Mutating these residues affect the EC50 of carboprost towards the FP receptor (supplementary Fig. 5b, c).

### Toggle switch residue difference influences prostaglandins' selectivity

One of the interesting phenomena is that apart from the FP receptor, PGF$_{2\alpha}$ and PGF$_{2\alpha}$ analogs show relatively higher affinity towards the EP3 receptor over the other prostaglandin receptors[22], this is also why the off-target side effects of carboprost are mainly due to non-specific activation of EP3 receptor. This also suggests that the binding pockets of the FP receptor and EP3 receptor share a certain level of similarity compared to other prostaglandin receptors. To understand this phenomenon, we compared our FP receptor structure with other agonist-bound active-state prostaglandin receptors, including EP2 receptor, EP3 receptor and EP4 receptor[5,6,27]. Through structure comparison, we notice FP receptor and EP3 receptor have more compact orthosteric pocket compared to the other two structures (Supplementary Fig. 4b). A close analysis of the residues interacting with the ligands reveals that the difference may come from the toggle switch residue, which is W$^{6.48}$ in both the FP receptor and EP3 receptor but is S$^{6.48}$ in EP2 receptor and EP4 receptor (Supplementary Fig. 4c, d). The bulkier side chain of tryptophan results in a more compact orthosteric pocket and stronger hydrophobic interaction with the ω-chain of the ligand. Indeed, mutating W$^{6.48}$ into S in both the FP receptor and EP3 receptor results in impaired potency towards carboprost or PGE$_2$, while mutating S$^{6.48}$ into W in EP4 receptor enhances its potency towards carboprost or PGE$_2$ (Supplementary Fig. 4e–g). However, the S$^{6.48}$W mutation has little effect on the EP2 receptor (Supplementary Fig. 4h), most likely because the neighboring L304$^{7.42}$ prevents the sidechain of W$^{6.48}$ to adopt the proper conformation. It should be noted that residue 7.42 is conserved as a smaller Ala in FP, EP3 and EP4 receptors

(Supplementary Fig. 4c). We then introduced L304[7.42]A mutation to EP2 receptor and checked the effect of S277[6.48]W mutation to the EP2_L304[7.42]A mutant. Indeed, both carboprost and PGE$_2$ show enhanced potency in the EP2_S277[6.48]W/L304[7.42]A compared to EP2_L304[7.42]A. (Supplementary Fig. 4i). The important role of the toggle switch W262[6.48] in ligand binding pose agrees with the previous study[5].

### The hydrogen bond network around the cyclopentane ring determines prostaglandins' selectivity

The most interesting phenomenon is that two endogenous ligands, PGE$_2$ and PGF$_{2\alpha}$, exhibit selectivity between the FP receptor and EP3 receptor, despite these two receptors sharing similar structures and activation mechanisms. As previously reported, PGF$_{2\alpha}$ has around 12-fold higher affinity towards the FP receptor compared to the EP3 receptor, while PGE$_2$ has around 360-fold higher affinity towards the EP3 receptor compared to the FP receptor[22]. Indeed, in a NanoBiT-based functional assay, we observed a similar profile of ligand potency (Supplementary Fig. 1).

The phenomenon is interesting because PGF$_{2\alpha}$ and PGE$_2$ share very similar chemical structures. The only difference comes from the 9-hydroxyl in PGF$_{2\alpha}$, which is a carbonyl in PGE$_2$ (Supplementary Fig. 7a). However, the differences between the carbon-oxygen single bond and double bond alter the shape of the cyclopentane ring and the electronegativity of the oxygen, which changes the chemical properties of the ligand (Supplementary Fig. 7b).

We used the FP receptor - carboprost structure to analyze the interaction between FP receptor and PGF$_{2\alpha}$, because carboprost and PGF$_{2\alpha}$ have identical cyclopentane ring. A close analysis of the sub-

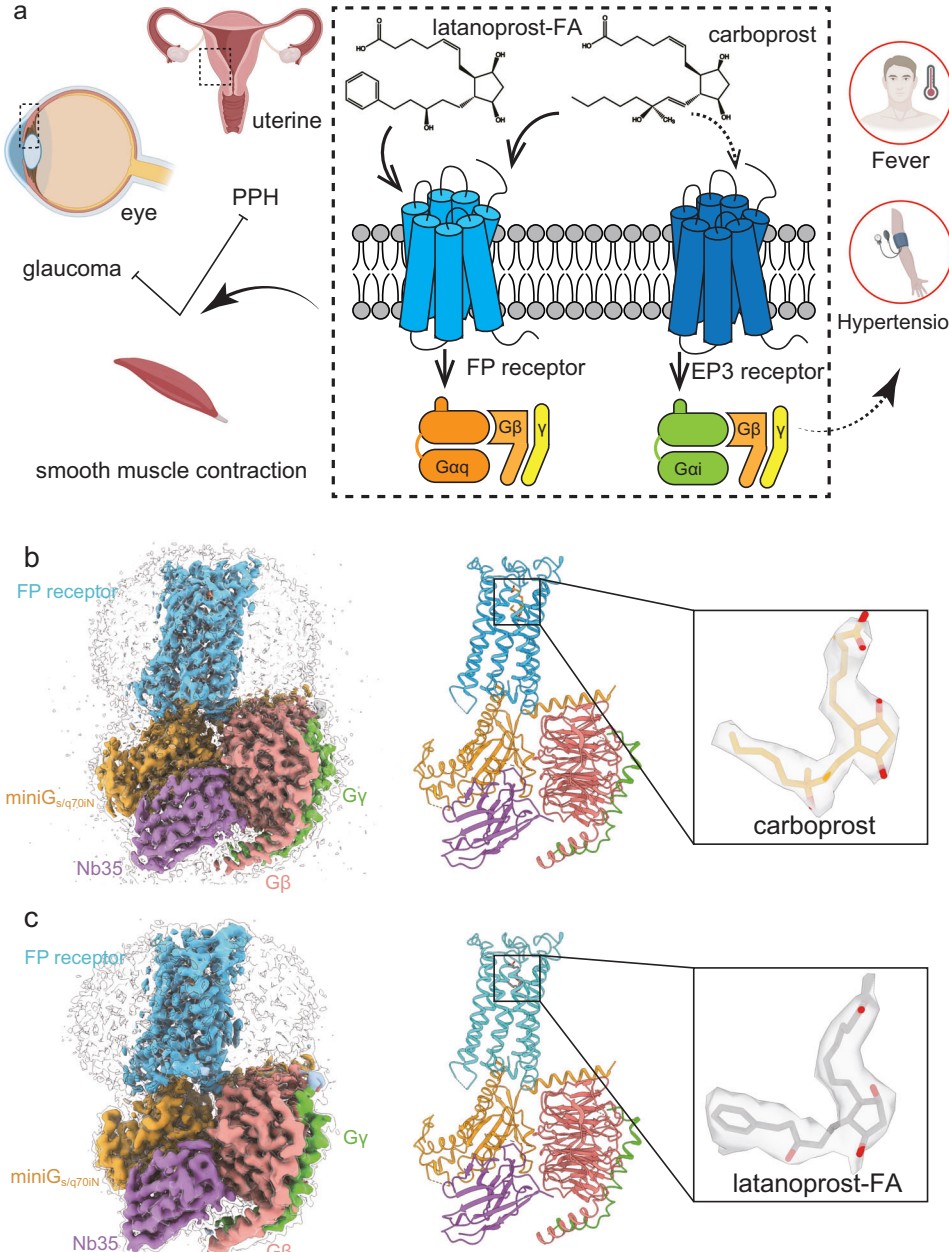

**Fig. 1 | The function of FP receptor and overall structures of FP-G$_{s/q70iN}$ complexes. a** Activation of the FP receptor promotes smooth muscle contraction. FP receptor agonists, such as latanoprost and carboprost, are clinically used to treat glaucoma and PPH. Off-target activation of the EP3 receptor by carboprost causes side effects such as hypertension and fever. Created with BioRender.com. The cryo-EM structures of FP-G$_{s/q70iN}$ in complex with carboprost (**b**) and latanoprost-FA (**c**). From left to right: the cryo-EM density of the entire complex, the structure coordinates of the complex and the electron density of carboprost or latanoprost-FA. Color code is as follows: FP receptor in blue, miniG$_{s/q70iN}$ in orange, Gβ in orange red, Gγ in green, Nb35 in purple, carboprost in orange, latanoprost-FA in gray.

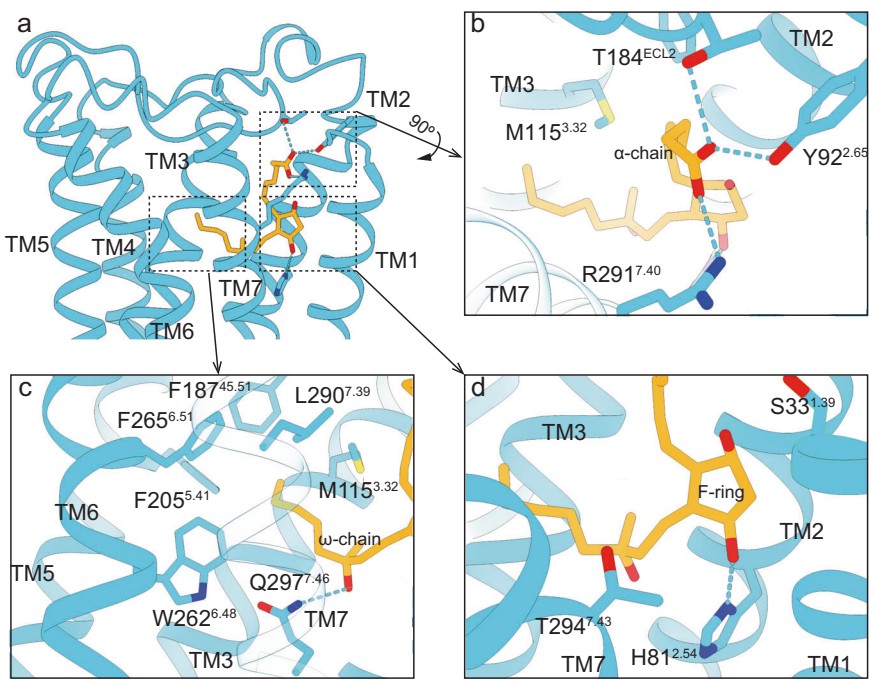

**Fig. 2 | The carboprost binding pocket. a** Overall view of the carboprost binding pocket. Carboprost adopts an L-shaped conformation and the binding pocket could be divided into three sub-pockets coordinating the α-chain (**b**), ω-chain (**c**), and F-ring (**d**) of the ligand. Hydrogen bonds are shown as blue dots.

pocket that interacts with the cyclopentane ring of the ligands reveals that several residues are not conserved in the FP receptor and EP3 receptor. These residues include S33[1.39], H81[2.54] and G85[2.58] of the FP receptor, which are P55[1.39], Q103[2.54] and T107[2.58] in the EP3 receptor. (Fig. 3a, b; Supplementary Fig. 8a).

In the FP receptor, S33[1.39] and T294[7.43] may interact with the 9-hydroxyl of PGF$_{2\alpha}$ through water-mediated hydrogen bonds. We could not confidently build the water molecules in the model due to resolution limitation, even though we observed weak densities potentially contributed by water molecules near the region. In the EP3 receptor, T294[7.43] is replaced with a similar residue S336[7.43], which maintains the ability to form hydrogen bonds. However, the S33[1.39] of the FP receptor is replaced with a P55[1.39] in the EP3 receptor, which loses the function of being a hydrogen bond donor or acceptor. Mutating S33[1.39] to P does not decrease the potency of PGF$_{2\alpha}$ but increases the potency of PGE$_2$ towards the FP receptor (Fig. 3c, d; Supplementary Fig. 8b; Supplementary Table 1). The results suggest that the extra space created by S33[1.39]P is important for PGE$_2$ binding, while the hydrogen bond between S33[1.39] and 9-hydroxyl of PGF$_{2\alpha}$ does not significantly contribute to the potency of the ligand, most likely because the interaction could be compensated by other hydrogen bonds between the F-ring and the receptor. Interestingly, the reverse P55[1.39]S mutations in the EP3 receptor only slightly decreased the potency for both PGE$_2$ and PGF$_{2\alpha}$ (Fig. 3e, f; Supplementary Fig. 8b; Supplementary Table 1). The results suggest other parts of the EP3 receptor also contribute to the high potency of PGE$_2$ to the receptor, and the side chain of serine is compatible with ligand binding.

Consistent with this, replacing G85[2.58] with threonine completely abolishes the ability of PGF$_{2\alpha}$ to activate the FP receptor, most likely because the space taken by the threonine side chain is incompatible with the cyclopentane ring (Fig. 3c, d; supplementary Fig. 7b). On the contrary, the T107[2.58]G mutation results in increased potency of PGF$_{2\alpha}$ towards the EP3 receptor, this again suggests the extra space to accommodate the F-ring is important (Fig. 3e, f; Supplementary Fig. 8b; Supplementary Table 1). As discussed before, the difference between the single bond and double bond on the cyclopentane ring of

the ligands not only changed the electro-negativity but also changed the shape and preferred orientation of the cyclopentane ring.

The 11-hydroxyl group of PGF$_{2\alpha}$ interacts with H81[2.54] in the FP receptor, while in the same position, the residue is Q103[2.54] in the EP3 receptor. Interestingly, switching the residues significantly impaired the binding of the ligands to the receptors. The FP_H81[2.54]Q mutation shows decreased potency to both PGF$_{2\alpha}$ and PGE$_2$, while the EP3_Q103[2.54]H mutation also shows decreased potency toward these two ligands (Fig. 3c–f; Supplementary Fig. 8b; Supplementary Table 1). The results suggest this residue may cooperate with nearby residues and contribute to the ligand binding. Replacing the residue disrupts the cooperation and results in a less favorable ligand binding pocket. Of note, all the mutations have comparable expression levels to the wildtype receptor (Supplementary Fig. 8c).

## Molecular mechanism of latanoprost-FA's high selectivity towards the FP receptor

As mentioned earlier, selective FP receptor agonists are preferred in PPH treatment to avoid "off-target" side effects. Latanoprost-FA represents the most selective FP receptor agonist, while carboprost is not as selective. Our NanoBiT-based G protein activation assay shows consistent results. As shown in Fig. 4, the EC50 of carboprost is similar in the FP receptor and EP3 receptor, while the EC50 of latanoprost-FA is around 1000-fold higher in the FP receptor than the EP3 receptor. Understanding the molecular mechanism of latanoprost-FA's high selectivity may guide the future development of better PPH drugs.

The cryo-EM map clearly reveals the binding pose of latanoprost-FA, which is very similar to that of carboprost. The benzene ring of latanoprost-FA fits into a hydrophobic pocket formed by F187[45.51], F205[5.41], F265[6.51] and W262[6.48] in the FP receptor (Fig. 4a, b). These residues are conserved in the EP3 receptor except for F265[6.51], which is a leucine in the EP3 receptor (Supplementary Fig. 6). Mutating F265[6.51] to leucine in the FP receptor significantly decreased latanoprost-FA's potency, while the reverse mutation L298[6.51]F in the EP3 receptor increased latanoprost-FA's potency (Fig. 4c, e; Supplementary Table 2). The results highlight the π-π interaction between F265[6.51] and the

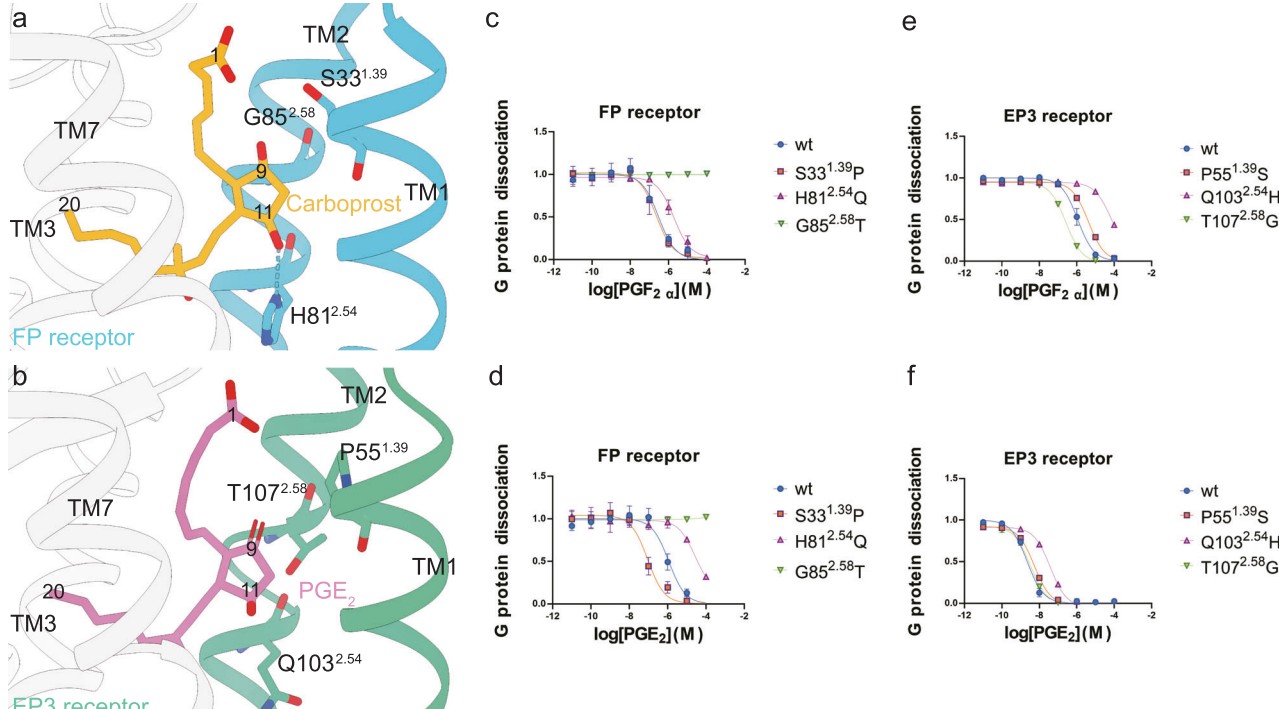

**Fig. 3 | Selectivity mechanism of PGF$_{2\alpha}$ and PGE$_2$ towards the FP receptor and EP3 receptor. a** H81$^{2.54}$ forms a hydrogen bond with 11-hydroxyl on the F-ring of carboprost (orange sticks) in the FP receptor (blue). The 9-hydroxyl interacts with S33$^{1.39}$ in the FP receptor in a space created by the small side chain of G85$^{2.58}$. **b** In the EP3 receptor (green), Q103$^{2.54}$ replaces H81$^{2.54}$ of the FP receptor and does not form a hydrogen bond with 11-hydroxyl of PGE$_2$ (red sticks). The 9-carbonyl of PGE$_2$ interacts with T107$^{2.58}$. S33$^{1.39}$ of the FP receptor is replaced by P55$^{1.39}$ in the EP3

receptor. **c** The PGF$_{2\alpha}$ activation profile of FP receptor and mutants revealed by NanoBiT assay. **d** The PGE$_2$ activation profile of FP receptor and mutants revealed by NanoBiT assay. **e** The PGF$_{2\alpha}$ activation profile of EP3 receptor and mutants revealed by NanoBiT assay. **f** The PGE$_2$ activation profile of EP3 receptor and mutants revealed by NanoBiT assay. Values represent the means ± SD of 3 independent experiments. Source data are provided as a Source Data file.

benzene ring of latanoprost-FA is the key to the selectivity. Of note, these mutations also slightly affect the potency of carboprost (Fig. 4d, e; Supplementary Table 2), but to a less extent compared to those of latanoprost-FA (Fig. 4e; Supplementary Table 2).

### Molecular mechanism of G-protein selectivity of prostaglandin receptors

Carboprost- and latanoprost-FA-bound FP-miniG$_{s/q70iN}$ complexes have almost identical G-protein binding interfaces. Thus, we use the carboprost-bound active state structure for the following structure analysis because the structure has a slightly higher resolution (Supplementary Fig. 9; Supplementary Fig. 10). As described above, miniG$_{s/q70iN}$ was used for structure determination. MiniG$_{s/q70iN}$ was engineered from miniGs by replacing its N-terminus with that of Gαi and replacing 7 residues in α5 helix with the equivalents of the Gαq. The mutations in α5 helix include R$^{H5.12}$K, Q$^{H5.16}$L, R$^{H5.17}$Q, H$^{H5.19}$N, Q$^{H5.22}$E, E$^{H5.24}$N, and L$^{H5.26}$V (superscript, CGN G protein numbering system[30], Fig. 5a). For this reason, structure analysis was only performed between the receptors and the α5 helix of G protein. Interestingly, the prostaglandin receptors EP2 and EP3 show different G protein coupling preferences, EP2 receptor couples to Gαs and EP3 receptor couples to Gαi[5,31]. The high-resolution structures of PGE$_2$-bound EP2-Gαs complex (PDB 7CX2) and PGE$_2$-bound EP3-Gαi complex (PDB 7WU9) have been reported[5,31]. We then performed structure comparison between these three structures to understand the G protein selectivity of the prostaglandin receptors.

Compared to EP2 and EP3 receptors, the TM6 of the FP receptor shows smaller outward displacement (Fig. 5b). The smaller outward shift of TM6 on the FP receptor results in a narrow G-protein binding pocket (Supplementary Fig. 11a), which leads to a different orientation of the α5 helix in the FP-miniG$_{s/q70iN}$ complex compared to the

EP2-Gαs complex and EP3-Gαi complex (Fig. 5b). The different α5 helix orientation is stabilized by salt bridge between R57$^{12.49}$ and the C-terminal carboxyl group of V$^{H5.26}$, as well as hydrogen bonds between H143$^{34.53}$ and N$^{H5.19}$, and between H244$^{6.30}$ and Q$^{H5.17}$ (Fig. 5c). Mutating these three residues affect the ability of FP receptor to activate G protein (Supplementary Fig. 11b, c). Interestingly, R57$^{12.49}$ and H143$^{34.53}$ are conserved in the Gq-coupling FP receptor and EP1 receptor, while H244$^{6.30}$ of FP receptor is D292$^{6.30}$ in EP1 receptor, which reserves the ability to interact with Q$^{H5.17}$. None of these three resides are conserved in the Gs-coupling EP2 receptor or Gi-coupling EP3 receptor (Fig. 5a). As a result, the specific orientation of α5 helix of Gαq could not be stabilized in the EP2 or EP3 receptors. Furthermore, N$^{H5.19}$ in the Gαq is H$^{H5.19}$ in Gαs, while Q$^{H5.17}$ in the Gαq is R$^{H5.17}$ in the Gαs and K$^{H5.17}$ in the Gαi (Fig. 5a). As a result, the FP receptor could not interact with Gαs or Gαi in the same manner as with the Gαq. Placing the α5 helix of the Gαs or Gαi in the intracellular central cavity of the FP receptor results in clashing between the α5 helix and the receptor (Fig. 5d, e). For example, H243$^{6.29}$ of the FP receptor may clash with R$^{H5.17}$ and R$^{H5.21}$ of Gαs (Fig. 5d), while H243$^{6.29}$ of the FP receptor may clash with F$^{H5.26}$ of Gαi (Fig. 5e). The extent of TM6 displacement is also considered as a determinant of G-protein preference in other receptors[32,33]. The aforementioned residues and the displacement of TM6 collectively contribute to the explanation of the preference of Gαq coupling for the FP receptor. It is also worth mentioning that the FP-miniG$_{s/q70iN}$ complex and the EP2-Gαs complex are mainly stabilized by electrostatic interactions between the receptor and the G protein, while the interactions between the EP3 receptor and Gαi are mainly contributed by hydrophobic contacts rather than electrostatic interactions (Supplementary Fig. 11d, e). This phenomenon was reported in other GPCR-G protein complexes[34].

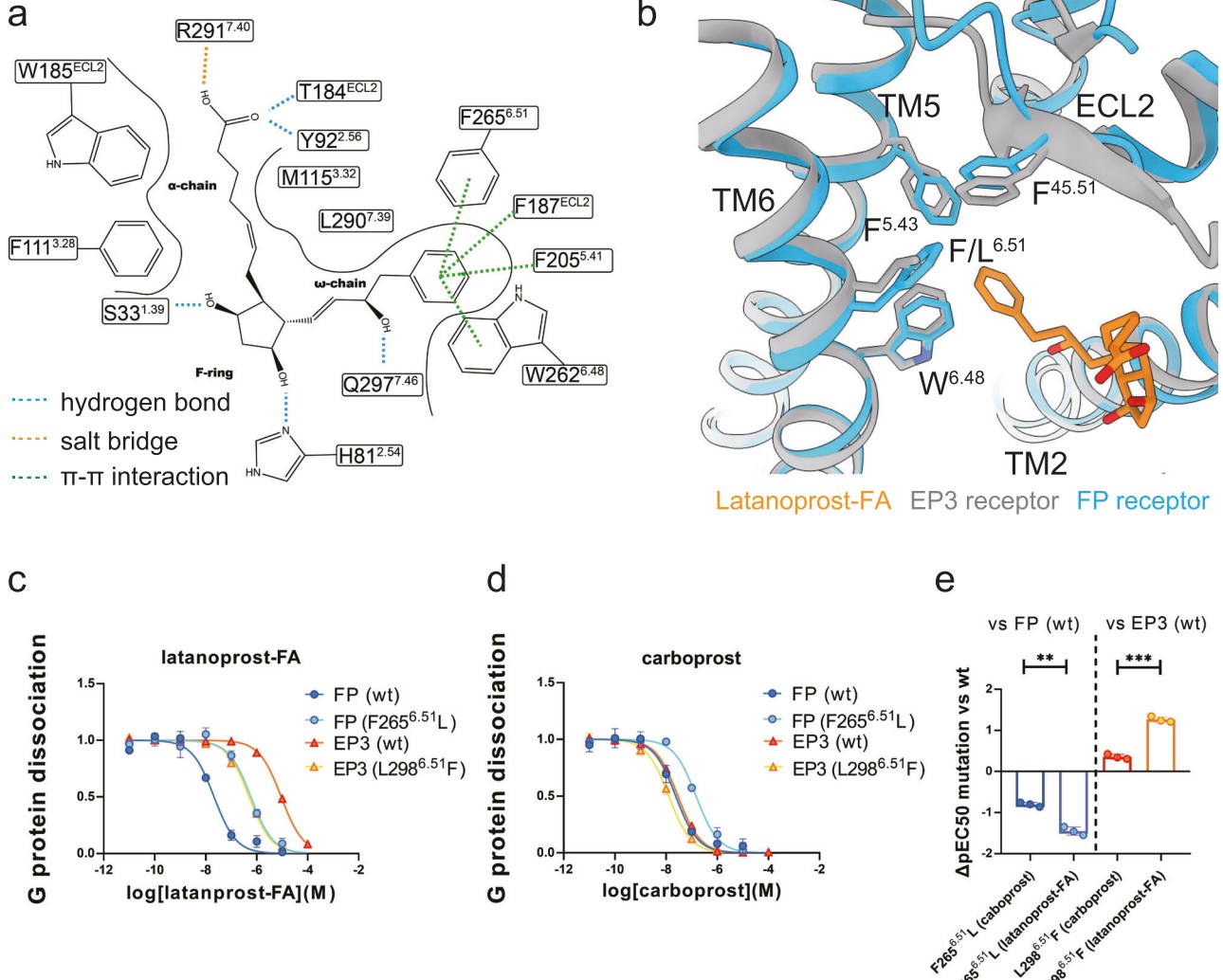

**Fig. 4 | Selectivity mechanism of latanoprost-FA toward FP receptor. a** Two-dimensional schematic depiction of the latanoprost-FA binding pocket on the FP receptor. H-bonds, salt bridges and π-π interactions are displayed as blue, orange and green dash lines respectively. Solid lines represent hydrophobic interfaces. **b** Comparison of the ω-chain binding sub-pocket between the FP receptor (blue) and EP3 receptor (gray). The only difference is at residue 6.51, which is a phenylalanine in the FP receptor and a leucine in the EP3 receptor. Latanoprost-FA (**c**) and carboprost (**d**)-mediated signaling on the wild-type FP, EP3 receptor, and FP (F265$^{6.51}$L) and EP3 (L298$^{6.51}$F) mutants. Values represent the means ± SD of 3 independent samples. **e** The ΔpEC50 of carboprost and latanoprost-FA bound to FP (F265$^{6.51}$L) and EP3 (L298$^{6.51}$F) from the wild-type receptors. Values represent the means ± SD of 3 independent samples. The significance of the value was determined using two tailed Student's $t$ test. **$P < 0.001$, ***$P < 0.0001$. Exact $p$ values and Source data are provided as a Source Data file.

## Discussion

Prostaglandins are a group of $C_{20}$ eicosanoids synthesized from the arachidonic acid metabolic pathway[35,36]. Different prostaglandin synthases produce four prostaglandins including $PGE_2$, $PGD_2$, $PGF_{2\alpha}$, and $PGI_2$, which activate different GPCRs. Except for $PGI_2$, the other prostaglandins share a similar chemical structure. Due to the critical roles of prostaglandins and their receptors in female reproduction, the analogs of prostaglandins, such as misoprostol and carboprost, have been widely used in clinical practice[37–39]. Carboprost is recommended by the International Federation of Gynecology and Obstetrics to treat PPH, but it has side effects caused by off-target activation of EP3 receptor[39]. In this work, we presented two cryo-EM structures of FP receptor bound with carboprost and latanoprost-FA. Structural analysis reveals how the FP receptors recognizes $PGF_{2\alpha}$ over the other chemically similar prostaglandins, and then mediates downstream physiological responses through the Gq subtype of G protein over the other subtypes. Our observations on the ligand recognition and G protein selectivity are further supported by a recent publication reporting the structures of FP receptor bound with $PGF_{2\alpha}$, TFPA and

LTPA[8] in complex with miniG$_{sqiN}$ and ScFv16. Apart from the slight conformational difference at the αN helix induced by ScFv16, the rest of the structures are very similar between our study and the published work. All the ligands bind to the same orthosteric pocket and adopt the "L-shaped" binding poses. Subtle differences are observed in the ligand conformation, which are likely due to imperfect model building limited by the resolution of the cryo-EM data. In all structures, the α5 helices of Gsq adopt similar conformation and the key interactions between the receptor and the G protein are well maintained.

Previous drug development of prostaglandin analogs generally focused on modifications on the α-chain or the ω-chain to improve pharmacokinetics properties or drug selectivity, such as acylation to α-carboxyl on the α-chain, introducing 15-methyl or 16-methyl or aromatic ring to the ω-chain[23,35,40,41]. Our structures as well as the mutagenesis studies suggest the sub-pocket interacting with the F-ring plays a key role in ligand selectivity. The high-resolution structure information provided in this work may guide drug development targeting on this sub-pocket for more FP receptor selective drugs.

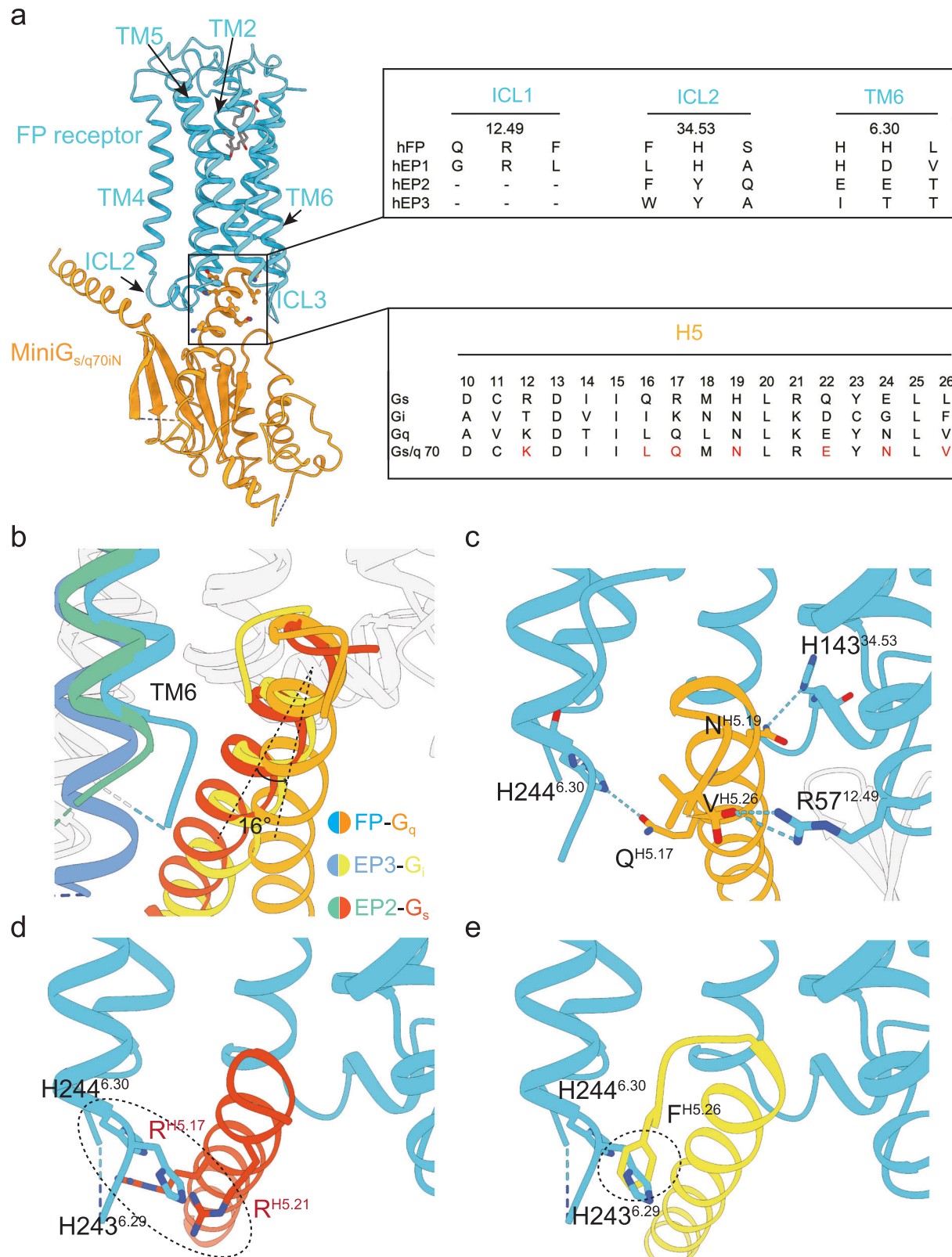

**Fig. 5 | Molecular mechanism of G-protein selectivity of prostaglandin receptors. a** Sequence alignments of key residues that determines the FP receptor – Gαq coupling specificity. **b** The TM6 displacement of the FP receptor, EP2 receptor and EP3 receptor when they couple with G protein. FP receptor, EP2 receptor and EP3 receptor are colored blue, green and deep blue respectively. Gαq, Gαs and Gαi are colored orange, red and yellow respectively. **c** The hydrogen bonds or salt bridge between FP receptor and Gαq (orange). **d** The potential clash between FP receptor and the α5 helix of Gαs (red) in a FP-Gαs model. **e** The potential clash between FP receptor and the α5 helix of Gαi (yellow) in a FP-Gαi model.

## Methods

### Protein construct design and cloning

The full-length FP receptor was cloned into the pFastBac vector with an N-terminal haemagglutinin (HA) signal peptide followed by a Flag-tag and C-terminal 6× His-tag to facilitate protein expression and purification. The miniG$_{s/q70iN}$ protein[26] was fused to the C terminus of FP receptor through a flexible glycine/serine linker (GGSGG) and rhinovirus 3 C protease recognition site (LEVLFQGP). MiniG$_{s/q70iN}$ was engineered from miniGs by replacing its N-terminus (residues 1-15) with that of Gαi (TLSAEDKAAVERSKM) and replacing 7 residues on the α5 helix with the equivalents of the Gαq (R$^{H5.12}$K, Q$^{H5.16}$L, R$^{H5.17}$Q, H$^{H5.19}$N, Q$^{H5.22}$E, E$^{H5.24}$N, and L$^{H5.26}$V).

For functional assays, we used the wild-type human FP receptor and human EP3 receptor (isoform A) with an N-terminal HA signal peptide and Flag-tag. To determine the expression level of wild-type FP receptor, EP3 receptor and their mutants, these GPCR receptors were inserted into pcDNA3.1 vector with the N-terminal haemagglutinin signal sequence followed by the Flag-tag and the HiBiT tag linked with flexible linkers. (MKTIIALSYIFCLVFA-DYKDDDDA-GGSGGGGSGGSSSG GG-VSGWRLFKKIS-GGSGGGGSGGSSSG).

### Protein expression and purification

Nanobody 35 (Nb35) with C-terminal 6×His-tag was cloned into the pET26b vector. The plasmid was transformed into BL21(DE3) *E. coli* cells and cultures in Terrific Broth (TB) medium. Cells were induced with 1 mM IPTG at an optical density (OD600) of 0.8 for 18 h at 20 °C and harvested by centrifugation. The LgBiT is expressed similarly except for using the pET22b vector instead of the pET26b vector.

The FP-miniG$_{s/q70iN}$ and Gβ$_1$γ$_2$ were cloned into the pFastBac vector, and baculoviruses were prepared using the Bac-to-Bac method. FP-miniG$_{s/q70iN}$ protein was expressed in Sf9 insect cells (Expression Systems, Cat # 94-0015) and Gβ$_1$γ$_2$ protein was expressed in Trichoplusia ni (Hi5) insect cells. For protein expression, insect cells at a density of around 3.0–4.0 × 10$^6$ cells per ml were transfected with baculoviruses and harvested after 48 h of infection. The cell pellets were stored at −80 °C until further use.

For the purification of Nb35, Gβ$_1$γ$_2$ and LgBiT, Ni-affinity chromatography was used as previously reported[25]. Briefly, the cell lysates containing the target protein were incubated with Ni-resin, the impurities were washed with buffer containing low concentration of imidazole (20 mM to 40 mM), while the target protein was eluted with buffer containing high concentration of imidazole (200 mM). The eluted Nb35 and LgBiT from Ni resin were further purified by size exclusion chromatography (SEC) while the eluted Gβ$_1$γ$_2$ was purified by reverse Ni-NTA affinity chromatography after being cleaved by HRV 3 C protease and dialyzed overnight.

For the purification of FP-miniG$_{s/q70iN}$ protein, cell pellets were lysed by resuspension in buffer containing 10 mM Tris HCl, pH 7.5, 1 mM EDTA, 20 μg/ml leupeptin, 160 μg/ml benzamidine, 8 mg/ml iodoacetamide and 1 μM carboprost tromethamine or latanoprost-FA. Cell membranes were extracted by centrifugation and washed twice with wash buffer (20 mM HEPES, pH 7.5, 300 mM NaCl, 100 μM TCEP and protease inhibitors). After homogenized in wash buffer, the membranes were incubated with 30 μg/ml Gβ$_1$γ$_2$, 10 μg/ml Nb35, 1 mM MnCl$_2$, 10 mM MgCl$_2$, and 50 μM carboprost tromethamine or latanoprost-FA overnight at 4 °C to reconstruct complexes. The complexes were extracted from cell membranes with solubilization buffer (20 mM HEPES, pH 7.5, 300 mM NaCl, 1% (w/v) DDM, 0.2% (w/v) cholesterol hemisuccinate (CHS), 10% glycerol, 10 μM carboprost tromethamine or latanoprost-FA, 10 μM TCEP and protease inhibitors) for 2 h at 4 °C. After high-speed centrifugation, the supernatant was loaded to M1 anti-Flag affinity resin. The detergent was gradually changed from DDM to 0.01% L-MNG on the M1 column. Finally, the protein was eluted with 20 mM HEPES, pH 7.5, 100 mM NaCl, 0.001% (w/v) L-MNG, 0.002% CHS, 10 μM TCEP, 5 mM EDTA and 0.2 mg/ml FLAG peptide.

Eluted FP-miniG$_{s/q70iN}$ was concentrated with a 50 kDa molecular weight cutoff (MWCO) spin concentrator to around 4 mg/mL.

The concentrated FP-miniG$_{s/q70iN}$ was mixed with Gβ$_1$γ$_2$ and Nb35 with a molar ratio of 1:1.2:1.2. The mixture was incubated on ice for 2 h and then purified by SEC in buffer containing 20 mM HEPES, pH 7.5, 100 mM NaCl, 0.001% (w/v) L-MNG, 0.00025% (w/v) GDN, 0.0002% CHS, 100 μM TCEP and 5 μM carboprost tromethamine or latanoprost-FA. The FP-miniG$_{s/q70iN}$/Gβ$_1$γ$_2$/Nb35 complex was concentrated with a 50 kDa MWCO spin concentrator to around 9 mg/ml for cryo-EM grids preparation.

### Cryo-EM grid preparation

For cryo-EM grid preparation, 3 μl of the purified carboprost tromethamine- or latanoprost-FA-bound FP-miniG$_{s/q70iN}$ complexes were applied onto glow-discharged Au Quantifoil grids. Grids were blotted with Whatman No. 1 qualitative filter paper in a Vitrobot Mark IV (Thermo Fisher) at 8 °C and 100% humidity for 4 s using a blot force of four before being plunged into liquid ethane.

### Data processing

Cryo-EM data were collected on a Titan Krios operating at 300 kV. Data processing was performed by cryoSPARC[42] (v3.1). 2, 664, 411 particles were picked from 1311 micrographs. After 2 rounds of 2D classification, a small subset of particles was used to generate an ab initial model. One good reference and two bad references generated from the initial model were used to perform 3 rounds of "guided multi-reference classification" as previously reported[43]. Briefly, Heterogeneous Refinement was used to perform the classification with the references volumes as input volumes[44]. Particles from the best class were merged while the other particles were removed, resulting in a subset of 327, 294 particles. After non-uniform refinement (NU-refine) and local refinement, the subset of particles was used to obtain the final map. This map has an indicated global resolution of 2.7 Å at a Fourier shell correlation of 0.143.

### Model building and refinement

The initial AlphaFold2 predicted model was generated from GPCRdb[45,46]. The coordinates of the NK1R-Gq complexes were used to generate the initial models of G$_{s/q70}$β1γ2 and Nb35 (PDB 7RMI). The coordinates and chemical restraints of carboprost and latanoprost-FA were generated using Phenix.elbow (1.20.1-4487)[47,48]. Models were initially docked into the density map by UCSF ChimeraX-1.3 and manually adjusted and rebuilt by COOT-0.9.8.7[49]. The structure refinement and validation were performed using PHENIX[47].

### NanoBiT G protein dissociation assay

Plasmids for the NanoBiT G protein dissociation assay were a general gift from Prof. Asuka Inoue. Cos7 cells (ATCC CRL-1651) were seeded in a six-well plate and allowed to grow to 50–80% confluence before transfection. A plasmid mixture containing 500–1500 ng LgBiT-inserted Gαq subunit, 500 ng Gβ$_1$, 500 ng SmBiT-fused Gγ$_2$ (C68S-mutant), 1000 ng resistance to inhibitors of cholinesterase-8A (Ric-8A) and 500–1500 ng FP receptor (or mutants) in 250 μl of Opti-MEM (Gibco) was transfected into cells using polyethylenimine (PEI) to measure Gq signaling. A plasmid mixture containing 400 ng LgBiT inserted Gαi subunit, 1000 ng Gβ1, 1000 ng SmBiT-fused Gγ2 (C68S), 2000 ng EP3 wild type and its mutation in 250 μl of Opti-MEM (Gibco) was transiently transfected with PEI to measure Gi signaling. After incubation for 24 h at 37 °C, transfected cells were harvested from the plate and resuspended in Hank's balanced salt solution (HBSS) (Gibco) with 20 mM HEPES, pH 7.5 and 10 μM coelenterazine 400a and then transferred into a 96-well plate. After one hour of incubation at room temperature, baseline luminescence was first measured. Different concentrations of ligands (10 μl) were then added and luminescence counts were measured every minute afterward. Data analysis was

performed using GraphPad Prism 9.1.1. Statistical analysis was performed using a two tailed Student's *t* test.

## Glo-Senor cAMP assay using an engineered Gsq protein

The Glo-Senor assay was used to evaluate GPCR-mediated cAMP accumulation. However, the FP receptor couples to Gq protein which could not activate adenylate cyclase and induce cAMP generation. Therefore, we constructed a Gsq chimera by replacing the last 15 amino acids of Gs with the last 15 amino acids of Gq. To perform the assay, Cos7 cells were seeded in a six-well plate and allowed to grow to 50–80% confluence before transfection. A plasmid mixture containing 100–500 ng FP receptor (or mutants), 250 ng Gsq, and 3750 ng pGloSensorTM-22F cAMP plasmid (Promega) in 250 μl of Opti-MEM (Gibco) was transfected into cells using PEI. 24 h after transfection, the transfected cells were harvested from the plate and resuspended in HBSS (Gibco) with 20 mM HEPES, pH 7.5 and 150 μg/ml luciferin and then transferred into a 96-well plate. After one hour of incubation at 37 °C followed by 1 h of incubation at room temperature, baseline luminescence was firstly measured. Different concentrations of ligands (10 μl) were then added and luminescence counts were measured every 2 min. Data analysis was performed using GraphPad Prism 9.1.1.

## HiBiT assay for expression level quantification

The receptors and their mutations were inserted into the pcDNA3.1 vector as above description. Cos7 cells were seeded in a six-well plate and allowed to grow to 60% confluence before transfection. The plasmids (1000 ng) in 250 μl of Opti-MEM (Gibco) were transfected into cells using PEI. After transfected for 24 h at 37 °C in 5% CO$_2$, Cos7 cells were plated on a 96-well plate using DMEM + 10%FBS. After 18 h, cells were washed with D-PBS to remove the complete medium and loaded with 45 μl of 10 μM coelenterazine diluted in HBSS plus 20 mM HEPES, pH 7.5 per well. The plate was measured for baseline luminescence. Then the LgBiT was added to each well and after 5 min, the plate was measured for second luminescence. The value of the luminescence presents the expression of the receptor.

## Cell surface staining

The expression levels of FP, EP2, EP3, EP4 receptors and mutants were conducted using cell surface staining. The transfected cells used for the functional assay were used for staining. In brief, the cells were resuspended in HBSS supplemented with 20 mM HEPES, pH 7.5 and incubated with Alexa-488 conjugated anti-Flag antibody (diluted with HBSS at a ratio of 1:300, Thermo Fisher, Cat # MA1-142-A488) in the dark for 15 min at room temperature. Cells were washed twice before the expression levels were detected by flow cytometry with excitation at 488 nm and emission at 519 nm. Standard gating strategy was applied based on the size and granularity of the cells (Supplementary Fig. 12).

## Statistics and reproducibility

A sample size of $n = 3$ is commonly used in biological studies. All experiments were performed in at least three independent biologial replicates. No data were excluded from the analyses. All attempts at replication were successful.

## Reporting summary

Further information on research design is available in the Nature Portfolio Reporting Summary linked to this article.

## Data availability

The atomic coordinates and the electron microscopy maps of FP-miniG$_{s/q70iN}$ complexes have been deposited in the Protein Data Bank (PDB) with the accessing code 8IQ4 (carboprost bound FP-miniG$_{s/q70iN}$-Gβ$_1$γ$_2$-Nb35 complex) and 8IQ6 (latanoprost-FA bound FP-miniG$_{s/q70iN}$-Gβ$_1$γ$_2$-Nb35 complex) as well as in the Electron Microscopy Data Bank

(EMDB) with the identification numbers EMD-35657 [https://www.ebi.ac.uk/pdbe/entry/emdb/EMD-35657] (carboprost bound FP-miniG$_{s/q70iN}$-Gβγ$_2$-Nb35 complex) and EMD-35658 [https://www.ebi.ac.uk/pdbe/entry/emdb/EMD-35658] (latanoprost-FA bound FP-miniG$_{s/q70iN}$-Gβ$_1$γ$_2$-Nb35 complex), respectively. Previously published structures can be accessed via accession codes: 6AK3 (PGE$_2$ bound EP3 receptor structure); 5YWY (ONO-AE3-208 bound EP4-Fab complex structure); 7CX2 (EP2-Gs complex structure); 7WU9 (EP3-Gi complex structure); 7D7M (EP4-Gs complex structure); 7RMI (NK1R-Gq complex structure). Source data are provided with this paper.

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

## Acknowledgements

We gratefully acknowledge the support from the Beijing Advanced Innovation Center for Structural Biology, Tsinghua University, Tsinghua-Peking Center for Life Sciences (CLS) (X.L.), by the National Natural Science Foundation of China (81974236 to W.Z. and X.L., Grant 32122041 to X.L.). We thank Prof. Asuka Inoue (Tohoku University) for providing the plasmids for the NanoBiT G protein dissociation assay.

## Author contributions

X.Lv., J.N., Y.R. and X.S. performed FP-miniGs/q70iN, Nb35 and Gβ1γ2 protein expression. X.Lv. prepared the FP-miniGs/q70iN-Gβ1γ2-Nb35 complexes. S.Z. prepared the cryo-EM sample. Xin.Z. and K.G. collected the cryo-EM data. K.G. and X.Lv. performed structure determination and refinement. K.G., X.Lv., J.N. and W.Z. characterized the biochemical properties of FP receptor, EP3 receptor and their mutants. Q.L., J.H., L.L. and X.Z. prepared the cDNA of FP receptor. X.L. and W.Z. coordinated the experiments and oversaw the overall study. The paper was written by X.Lv., K.G., W.Z. and X.L. with input from J.N. All authors contributed to the editing of the paper.

## Competing interests

The authors declare no competing interests.
