## [Peer Review File · Nature Communications]

Prostaglandin F₂ α receptor structures reveal the mechanism of ligand and G protein selectivityReviewer #1 (Remarks to the Author):

Lv et al report an important study to illustrate the molecular mechanisms of the ligand selectivity and G protein selectivity of the prostaglandin F2 alpha (FP) receptor using cryo-EM studies and functional assays. An agonist of FP receptor, carboprost has been used for the treatment of postpartum hemorrhage. However, its activation of the EP3 receptor leads to severe side effects and limits its clinical application. In contrast, latanoprost shows remarkably higher selectivity on the FP receptor than the EP3 receptor. To understand how the ligand selectivity is achieved, the authors determined two cryo-EM structures of the FP-miniGs/q70iN protein complex bound to carboprost or latanoprost. These structures also provide a possible explanation for the selectivity of the FP receptor for the endogenous agonist PGF₂α and Gq. The manuscript is well written. Below are major and minor points the authors should address before its acceptance.

Major:

1. The title should be revised. Abbreviated name is not recommended in the title. FP receptor should be changed to prostaglandin F₂α receptor. This paper reveals molecular mechanisms of receptor selectivity for both prostaglandin and clinical drugs. Simply emphasizing the selectivity mechanism of prostaglandins is not accurate.
2. The authors should include the introduction of all four prostaglandin E2 receptors and their G protein selectivity as well as previous structural studies of these receptors in the introduction part since they are closely related.
3. The authors claim H244, R57 and H143 of FP receptor is important for determining Gq coupling. Mutagenesis studies using NanoBiT assay is highly encouraged to support this conclusion. The EP1 receptor also couples to Gq. Sequence alignment of all four EP receptors and FP would be helpful to understand the G protein selectivity.

Minor:

1. The NanoBiT-based data can not simply reflect the binding affinity of ligands. For example, antagonists have high binding affinity for the receptor but fail to activate the receptor. Thus, "affinity" should be changed to potency throughout the paper.
2. Character size in some figures is too small. In Figure 3 a, and b, the receptors and ligands name should be labeled to facilitate reading. The double bond of the 9-carbonyl group of PGE2 can be shown to differentiate it from the hydroxyl group of PGF₂α.
3. Side chains of G852.58 and T1072.58 can be shown in supplementary Fig. 4b to see if T1072.58 can clash with PGF₂α.
4. The authors should show the complete structure of latanoprost in Fig 4b.
5. line 99, contributed should be contributes.
6. Line 181-183, the meaning is not clear. This sentence can be changed to "replacing G85 with threonine completely abolishes the ability of PGF₂α to activate the FP receptor"
7. line 120, influents should be influences.
8. line 122, high should be higher
9. line 257, considered as.
10. Cell dimensions parameters in Supplementary Table 3 should be removed.

Reviewer #2 (Remarks to the Author):

In the manuscript entitled "FP receptor structures reveal the selectivity mechanism of prostaglandins" by Liu and colleagues, the authors determined the CryoEM structures of prostaglandin F receptor (FP) in complex with engineered Gq protein and two different agonists, carboprost and latanoprost free acid. The FP receptor is a target for the treatment of glaucoma and postpartum hemorrhage, making the structural information of the FP receptor highly valuable. This study provides a valuable addition to the previous report by Wu et al. entitled "Ligand-induced activation and G protein coupling of prostaglandin F₂α receptor".

There are several differences between the two reports: Lv et al. did not use scFv16 or NanoBiT for stabilization, while Wu et al. did. Additionally, Lv et al. used 15-methyl PGF₂α (carboprost) instead of PGF₂α used by Wu. Since 15-methylation is a representative

strategy to make PG resistant to metabolism, the comparison of PGF_{2α} and 15-methyl PGF_{2α} provides important information.

While the EP3 receptor is the PG receptor most closely related to the FP receptor, Lv et al. discussed the molecular mechanism of selectivity of PGF_{2α} and latanoprost through structural comparison and mutagenesis experiments. Furthermore, they identified several unique residues involved in Gq selectivity.

Overall, this study provides useful information for further pharmacological analysis and drug development and will find broad interest in the GPCR field. Some important caveats are highlighted below, that need to be addressed.

Major comments:

- 1) The journal Nature Communications targets a wide range of researchers across various fields, and therefore, a more general introduction is necessary. The current introduction of the manuscript is limited to FP and EP3, and there seems to be an insufficient explanation regarding other PG receptors, their ligands, signaling pathways, and the structures solved thus far.
- 2) It is mentioned in line 96 that ECL2 forms a hairpin structure, which is an important common feature of PG receptors and should be shown in the figure. Additionally, there appears to be a lack of mutagenesis experiments on important residues for ligand recognition, such as Y92^{2.65}, T184^{ECL2}, and R291^{7.40}.
- 3) In Fig. 3, although functional validation was performed using PGF_{2α} instead of carboprost, it should be verified whether the same results are obtained when using carboprost. In particular, the 11-hydroxyl group is recognized by T294^{7.43} in Wu et al.'s report, and it is an important issue whether this difference is responsible for the difference between PGF_{2α} and 15-methyl PGF_{2α}. While 11-deoxy PGF_{2α} is commercially available, verifying which residue recognizes the 11-hydroxyl group may be possible by examining the activation of the H81^{2.54} and T294^{7.43} mutants by 11-deoxy PGF_{2α}.
- 3) On lines 131 to 134, the authors suggest that the higher cross-reactivity of EP3 with PGF_{2α} compared to EP2 and EP4 may be due to the presence of W at position 6.48 in EP3 and FP, while EP2 and EP4 have S at this position. While the authors have created EP3 and FP W6.48S mutants and demonstrated the contribution of W^{6.48} to the recognition of PGE₂ and PGF_{2α}, these mutants still maintain their selectivity for PGE₂ or PGF_{2α}. On the other hand, it would be informative to also test whether the reverse mutation, S6.48W in EP2 and EP4, enhances their affinity for PGE₂ and PGF_{2α}.
- 4) In line 192, the statement "switching the residues significantly impaired the binding of the ligands to the receptors" is not entirely accurate, as the signaling assays used in the study only evaluate receptor activation, not ligand binding. Similarly, the term "affinity" is used in several instances throughout the manuscript (e.g. lines 137, 149, 170, 177, 193), which specifically refers to binding, while only receptor activity was assessed in the experiments. Thus, it is important to be mindful of the terminology used when describing the result
- 5) In Fig. 5, the importance of three interactions for Gq selectivity is noted. However, the contribution of each interaction should be verified through mutagenesis experiments. Specifically, the authors have the capability to conduct NanoBiT G protein dissociation assays, which would undoubtedly strengthen their argument by investigating the effects of mutations in the Gq protein.
- 6) Wu et al. also determined the structures of the FP receptor bound to PGF_{2α} and latanoprost free acid. If possible, it would be beneficial to compare and discuss whether there are any differences between their structures and the structures determined by the authors.

Minor points:

- Line 14: No. 5 affiliation is not included in the list of authors.
- Line 25: "female production" seems to be a typo for "female reproduction."
- Line 48: While it may be an editorial convenience, the final version should correctly subscript 2 α in PGF_{2 α} .
- Line 52: Carboprost tromethamine is described as a mixture of 15-methyl PGF_{2 α} and tromethamine, but I think "salt" may be a more accurate term.
- Line 62: To my knowledge, the paper cited in reference 12 only shows the affinity of PGF_{2 α} , not carboprost specifically. Could you please indicate where carboprost is mentioned in the paper?
- Line 80: The notations miniG_{s/q70iN} and miniG_{sqiN} are used multiple times in the manuscript. If they refer to the same thing, they should be unified.
- Lines 80 to 83: The authors should first mention the difference between the miniG_{s/q70} used in ref.14 and ref.15, which does not have the N-terminus replaced by Gi, and the miniG_{s/q70iN} used in their study. In this section, it should also be noted that only the 7 residues in the α 5 helix were replaced with those from Gq. Furthermore, the Materials and Methods section should clearly state the precise sequence of the substitution.
- Lines 146 to 147: In reference 12, Table 2 shows the Ki values for PGF_{2 α} towards EP3 and FP as 38 nM and 3.2 nM, respectively, and the Ki values for PGE₂ towards EP3 and FP as 0.33 nM and 119 nM, respectively. Based on these values, PGF_{2 α} and PGE₂ appear to have selectivity values of 12-fold and 360-fold, respectively. However, there is a discrepancy between the selectivity values mentioned by the authors and my calculated results. Can you please clarify which values were used for the authors' calculations?
- Line 209: As noted by the authors, "latanoprost" specifically refers to the isopropyl ester form, which is a prodrug. Therefore, it is recommended to use a different term such as "latanoprost-FA" to refer to the free acid form.
- Line 378: The correct full name of Ric-8A is resistance to inhibitors of cholinesterase-8A.
- Line 592: The authors with the same initials X.Z. should be represented in a way that allows for a clear distinction between the two individuals.
- Line 665: The manuscript mentions "3 independent experiments", but in Supplementary Fig. 5c, there appear to be around 8 dots.
- Line 704: "hydrophic" seems to be a typo for "hydrophilic."
- Fig. 1a: The arrow between "uterine" and "PPH" is unclear in its meaning. Could you please provide an explanation?
- Fig. 1b: The label/number for 'b' is missing in the figure.
- Fig. 2c-h, Supplementary Fig. 2c: It is confusing because it appears that activation is suppressed as the ligand concentration increases. Consider changing the Y-axis to G protein dissociation or setting the activation state value to 1.
- Fig. 4a: The double bond between positions 5 and 6 in latanoprost-FA is in the cis configuration.
- Supplementary Fig. 3: The residues before and after 5.41/5.43 are reversed in all receptors.

- Supplementary Fig. 4a: The stereochemistry at position 15 of PGE₂ and PGF_{2α} should be indicated for clarity.
- Supplementary Fig. 7d, e: To facilitate comparison of carboprost-bound and latanoprost-bound forms, they should be shown as oriented as possible. For example, Y92^{2.65} and W262^{6.48} are visible in d but not in e.
- Supplementary Tables. 1, 2: pEC50 is defined as the negative common logarithm of EC50, so the signs of the values in Supplementary Table 1 are expected to be opposite to what is shown. Similarly, the vertical axis of Fig. 4e and Supplementary Fig. 5b appears to be inverted.
- A schematic representation of carboprost recognition, similar to Fig. 4a, should also be included for clarity.

Reviewer #3 (Remarks to the Author):

Postpartum hemorrhage (PPH) accounts for approximately 18% of all deaths of pregnant women globally and represents a leading cause of maternal morbidity and mortality. Carboprost tromethamine is the most effective treatment of atonic PPH by activating the prostaglandin F₂-alpha receptor (FP receptor), however, it has side effects such as fever and hypertension due to its activation of the closely related EP₃ receptor. The off-target side effects limit its clinical application, especially to patients with cardiovascular diseases.

In this manuscript, Lv et al. reported the high-resolution cryo-EM structures of the human prostaglandin F₂-alpha receptor (FP receptor) bound with carboprost or a more selective drug latanoprost. The structural information reveals the molecular mechanism underlying the FP receptor selectivity of these clinical drugs and provides guidance for the development of better PPH drugs. The work is of general interest to the field.

I have the following minor suggestions:

1. The label for panel b is missing in figure 1.
2. The labeling in Figure 5a is too small. The authors should adjust the size of the labelling.
3. The cryo-EM data processing flow-chart (supplementary figure 7) lacks the essential details. For example, how many particles were used to calculate the final maps for each structure? Also in panel d and e, the readers may understand these are density maps for TM1-7, but it would be more clear if these helices are labelled. Please show densities of the residues in orthosteric site, H8 helix, and extra- or intra-cellular loops.
4. The authors may consider using the template provided by Nature to prepare the cryo-EM data collection and processing table. (<https://www.nature.com/nature/for-authors/formatting-guide>)

We appreciate the suggestions from the reviewers and have addressed all the comments. The point-by-point responses to the concerns/suggestions are provided below in blue font.

Reviewer #1 (Remarks to the Author):

Lv et al report an important study to illustrate the molecular mechanisms of the ligand selectivity and G protein selectivity of the prostaglandin F2 alpha (FP) receptor using cryo-EM studies and functional assays. An agonist of FP receptor, carboprost has been used for the treatment of postpartum hemorrhage. However, its activation of the EP3 receptor leads to severe side effects and limits its clinical application. In contrast, latanoprost shows remarkably higher selectivity on the FP receptor than the EP3 receptor. To understand how the ligand selectivity is achieved, the authors determined two cryo-EM structures of the FP-miniG_{s/q70IN} protein complex bound to carboprost or latanoprost. These structures also provide a possible explanation for the selectivity of the FP receptor for the endogenous agonist PGF_{2α} and Gq. The manuscript is well written. Below are major and minor points the authors should address before its acceptance.

We thank the reviewer for the positive comments on our work.

Major:

1. The title should be revised. Abbreviated name is not recommended in the title. FP receptor should be changed to prostaglandin F_{2α} receptor. This paper reveals molecular mechanisms of receptor selectivity for both prostaglandin and clinical drugs. Simply emphasizing the selectivity mechanism of prostaglandins is not accurate.

We thank the reviewer for the suggestion. We have changed the title to 'Prostaglandin F_{2α} receptor structures reveal the mechanism of ligand and G protein selectivity' in the revised manuscript.

2. The authors should include the introduction of all four prostaglandin E2 receptors and their G protein selectivity as well as previous structural studies of these receptors in the introduction part since they are closely related.

We agree with the reviewer for the suggestion and have added the introduction of other prostaglandin receptors at the introduction session.

3. The authors claim H244, R57 and H143 of FP receptor is important for determining Gq coupling. Mutagenesis studies using NanoBiT assay is highly encouraged to support this conclusion. The EP1 receptor also couples to Gq. Sequence alignment of all four EP receptors and FP would be helpful to understand the G protein selectivity.

We thank the reviewer for the suggestion. We performed the mutagenesis studies (R57A, H143A, H244A and H244F) as the reviewer suggested using NanoBiT assay (Response Figure 1a) and GloSensor cAMP assay (Response Figure 1b). We don't see obvious effects of these mutations with NanoBiT assay, likely because the NanoBiT assay has its limitations of short detection window and relatively low signal amplification (thus low sensitivity) as the G protein subunits undergoes dissociation and association during the time. We then used GloSensor cAMP assay which may have higher sensitivity due to the

amplification of the signals. Indeed, we observed decreased Emax of the mutations compared to the wt FP receptor, even though the EC50 values are similar. It should be noted that in the recently published FP structure paper, Wu. et al reported similar effect of H244A mutation (similar EC50, decreased Emax) using IP-1 assay. The GloSensor cAMP assay results are now included in the supplementary figure 10 of revised manuscript. We also added EP1 alignment in Figure 5a, interestingly, hEP1 also has R^{12.49} and H^{34.53}. However, it has D^{6.30} instead of H^{6.30}. As “D^{6.30}” is also able to interact with Q^{H5.17}, such amino acid difference might be tolerated.

Response Figure 1. The mutagenesis studies of R57^{12.49}A, H143^{34.53}A, H244^{6.30}A and H244^{6.30}F. **a** The carboprost activation profiles of FP receptor and mutants revealed by NanoBiT assay. **b** The carboprost activation profiles of FP receptor and mutants revealed by GloSensor cAMP assay. **c** The expression levels of the R57^{12.49}A, H143^{34.53}A, H244^{6.30}A, H244^{6.30}F mutants and wide type FP receptor measured by cell surface staining. Data are given as mean \pm SD from 3 independent experiments.

Minor:

1. The NanoBiT-based data can not simply reflect the binding affinity of ligands. For example, antagonists have high binding affinity for the receptor but fail to activate the receptor. Thus, “affinity” should be changed to potency throughout the paper.

We agree with the reviewer and have changed the term ‘affinity’ to ‘potency’ as suggested.

2. Character size in some figures is too small. In Figure 3 a, and b, the receptors and ligands name should be labeled to facilitate reading. The double bond of the 9-carbonyl group of PGE₂ can be shown to differentiate it from the hydroxyl group of PGF_{2 α} .

We have adjusted the character size and added the receptors and ligands name in Figure 3 a, and b. We have shown the double bond of the 9-carbonyl group of PGE₂ in Figure 3 in the revised manuscript.

3. Side chains of G85^{2.54} and T107^{2.58} can be shown in supplementary Fig. 4b to see if T107^{2.58} can clash with PGF_{2 α} .

We thank the reviewer for the suggestion. We showed the side chains of G85^{2.58} and T107^{2.58} in the supplementary figure 7b of revised manuscript (response figure 2). The sidechain of T107^{2.58} is close to the F-ring of carboprost and there is likely clash between them. As shown in response figure 6, PGF_{2 α} and carboprost behave similarly in the functional assay. Thus T107^{2.58} may also clash with the F-ring of PGF_{2 α} .

Response Figure 2. The sidechain of T107^{2.58} is close to the F-ring of carboprost and there is likely clash between them.

4. The authors should show the complete structure of latanoprost in Fig 4b.
We have showed the complete structure of latanoprost in Fig 4b (Response figure 3).

Response figure 3. The complete structure of latanoprost.

5. line 99, contributed should be contributes.
We have corrected 'contributed' to 'contributes'.

6. Line 181-183, the meaning is not clear. This sentence can be changed to "replacing G85 with threonine completely abolishes the ability of PGF_{2α} to activate the FP receptor"
We have adjusted the sentence as the reviewer suggested.

7. line 120, influents should be influences.

We have corrected 'influents' to 'influences'.

8. line 122, high should be higher.

We have corrected 'high' to 'higher'.

9. line 257, considered as.

We have corrected 'considered' to 'considered as'.

10. Cell dimensions parameters in Supplementary Table 3 should be removed.

We have removed these parameters in Supplementary Table 3.

Reviewer #2 (Remarks to the Author):

In the manuscript entitled "FP receptor structures reveal the selectivity mechanism of prostaglandins" by Liu and colleagues, the authors determined the CryoEM structures of prostaglandin F receptor (FP) in complex with engineered Gq protein and two different agonists, carboprost and latanoprost free acid. The FP receptor is a target for the treatment of glaucoma and postpartum hemorrhage, making the structural information of the FP receptor highly valuable. This study provides a valuable addition to the previous report by Wu et al. entitled "Ligand-induced activation and G protein coupling of prostaglandin F_{2α} receptor".

There are several differences between the two reports: Lv et al. did not use scFv16 or NanoBiT for stabilization, while Wu et al. did. Additionally, Lv et al. used 15-methyl PGF_{2α} (carboprost) instead of PGF_{2α} used by Wu. Since 15-methylation is a representative strategy to make PG resistant to metabolism, the comparison of PGF_{2α} and 15-methyl PGF_{2α} provides important information.

While the EP3 receptor is the PG receptor most closely related to the FP receptor, Lv et al. discussed the molecular mechanism of selectivity of PGF_{2α} and latanoprost through structural comparison and mutagenesis experiments. Furthermore, they identified several unique residues involved in Gq selectivity.

Overall, this study provides useful information for further pharmacological analysis and drug development and will find broad interest in the GPCR field. Some important caveats are highlighted below, that need to be addressed.

We thank the reviewer for the positive comments on our work.

Major comments:

1) The journal Nature Communications targets a wide range of researchers across various fields, and therefore, a more general introduction is necessary. The current introduction of the manuscript is limited to FP and EP3, and there seems to be an insufficient explanation regarding other PG receptors, their ligands, signaling pathways, and the structures solved thus far.

We have added the introduction of other prostaglandin receptors in the introduction part.

2) It is mentioned in line 96 that ECL2 forms a hairpin structure, which is an important common feature of PG receptors and should be shown in the figure. Additionally, there appears to be a lack of mutagenesis experiments on important residues for ligand recognition, such as Y92^{2.65}, T184^{ECL2}, and R291^{7.40}.

We thank the reviewer for the helpful suggestions. We prepared a new figure to show the hairpin structure of ECL2 from prostaglandin receptors in the revised manuscript (Revised supplementary figure 4a; Response figure 4). We also performed mutagenesis studies of important residues on ligand recognition, including Y92^{2.65}A, T184^{ECL2}A, R291^{7.40}A, as well as H81^{2.54}A and T294^{7.43}A. As shown in response figure 5 (Supplementary Figure 5 in the revised manuscript), these mutations affect the EC₅₀ of carboprost compared to FP (wt). The results are consistent with previous reports (Wu et al's, Nature communications).

Response figure 4. Hairpin structure formed by ECL2 (red) in prostaglandin receptors.

Response Figure 5. Mutagenesis studies on the key residues for ligand recognition.

a The carboprost activation profiles of FP receptor and the binding pocket mutants revealed by NanoBiT assay. **b** The pEC₅₀ of wt FP receptor and the binding pocket mutants as well the ΔpEC₅₀ of these mutants from wt FP receptor.

3) In Fig. 3, although functional validation was performed using PGF_{2α} instead of carboprost, it should be verified whether the same results are obtained when using carboprost. In particular, the 11-hydroxyl group is recognized by T294^{7.43} in Wu et al.'s report, and it is an important issue whether this difference is responsible for the difference between PGF_{2α} and 15-methyl PGF_{2α}. While 11-deoxy PGF_{2α} is commercially available, verifying which residue recognizes the 11-hydroxyl group may be possible by examining the activation of the H81^{2.54} and T294^{7.43} mutants by 11-deoxy PGF_{2α}.

We thank the reviewer for the suggestion. As shown in response figure 6, $\text{PGF}_{2\alpha}$ and carboprost behave similarly in the functional assay.

It's interesting that our structure and Wu et al.'s structure suggest slightly different interaction pattern between the 11-hydroxyl group and the receptor. It should be noted that these two structures have very similar overall conformation but slightly different ligand binding poses. A close comparison between these two datasets suggests our structure has slightly higher local resolution at the ligand binding site. A comparison of the ligand densities as well as the key residue densities are shown in response figure 7. Notably, the modeling of T294^{7.43} in Wu et al.'s structure does not seem to fit the density very well (Response figure 7g). Of course, structure modeling is never perfect due to the limitation of electron densities. We also noticed that the different conformation of S33^{1.39} directly affects the F-ring of the ligand, while our structure and Wu et al.'s structure has slightly different conformation at the N-terminus. We want to point out that Wu et al used a N-terminal BRIL fusion construct of FP receptor while we did not use BRIL fusion, this difference might also contribute to the N-terminus dynamics and results in the local differences.

We agree with the reviewer that 11-deoxyl $\text{PGF}_{2\alpha}$ would be an ideal tool compound to check if H81^{2.54} or T294^{7.43} directly interact with 11-hydroxyl. We tried to purchase the compound for the past three months. Even though the vendor initially promised to provide the compound within 1 month, they eventually failed to find this compound on the market.

a

Response Figure 6. $\text{PGF}_{2\alpha}$ and carboprost behave similarly in the functional assay.

a The carboprost and $\text{PGF}_{2\alpha}$ activation profiles of FP receptor and mutants revealed by GloSensor cAMP assay. **b** The carboprost and $\text{PGF}_{2\alpha}$ activation profiles of EP3 receptor and mutants revealed by NanoBiT assay.

Response Figure 7. The comparison of electron densities for the ligand and key pocket residues in our FP-carboprost structure and the reported FP-PGF_{2α} structure. **a-d** The electron density of carboprost (a), H81^{2.54} (b), T294^{7.43} (c) and S33^{1.39} (d) in our structure. The ligand density is more continuous in our structure (a) compared to Wu et al.'s structure (e). **e-h** The electron density of PGF_{2α} (e), H81^{2.54} (f), T294^{7.43} (g) and S33^{1.39} (h) in the reported FP-PGF_{2α} structure. The red arrows in panel (g) indicate the modeling of T294^{7.43} does not fit the density well. **i** Overlay of our model and the reported FP-PGF_{2α} structure highlighting our carboprost model fits well with Wu et al.'s electron density (light yellow). **j** Overlay of our ligand model to the ligand density in Wu et al.'s structure. **k** Zoom-in view of the ligand density in Wu et al.'s structure.

4) On lines 131 to 134, the authors suggest that the higher cross-reactivity of EP3 with PGF_{2α} compared to EP2 and EP4 may be due to the presence of W at position 6.48 in EP3 and FP, while EP2 and EP4 have S at this position. While the authors have created EP3 and FP W^{6.48}S mutants and demonstrated the contribution of W^{6.48} to the recognition of PGE₂ and PGF_{2α}, these mutants still maintain their selectivity for PGE₂ or PGF_{2α}. On the other hand, it would be informative to also test whether the reverse mutation, S^{6.48}W in EP2 and EP4, enhances their affinity for PGE₂ and PGF_{2α}.

We thank the reviewer for the helpful suggestion. We checked the S^{6.48}W mutation in EP2 and EP4 receptors. The potency of PGF_{2α} is too low to obtain reasonable activation curves for EP2 and EP4 receptors, therefore we used the carboprost instead. Mutating S^{6.48} into W in EP4 receptor enhances potency towards carboprost or PGE₂ (Supplementary Fig. 4g; Response figure 8a). However, the S^{6.48}W mutation has little effect on the EP2 receptor (Supplementary Fig. 4h; Response figure 8b). We reasoned that a larger sidechain of L304^{7.42} in EP2 receptor prevents the correct configuration of W^{6.48}. Of note, residue 7.42 is conserved as a smaller Ala in FP, EP3 and EP4 receptors (Supplementary Fig. 4c; Response figure 8c). We then introduced L304^{7.42}A mutation to EP2 receptor and checked the effect of S277^{6.48}W mutation to the EP2_L304^{7.42}A mutant. Indeed, both carboprost and PGE₂ show enhanced potency in the

EP2_S277^{6.48}W/L304^{7.42}A compared to EP2_L304^{7.42}A. (Supplementary Fig. 4i; Response figure 8b). Taken together, the results support our statement that S/W difference contributes to the ligand potency differences in the FP, EP3, EP2 and EP4 receptors.

Response figure 8. The effect of S^{6.48} in EP2 and EP4 receptor for ligands' potency. a, b Carboprost and PGE₂-mediated signaling on EP4 receptor and EP4_S285^{6.48}W (a), EP2 receptor and EP2_S277^{6.48}W as well as EP2_L304^{7.42}A and EP2_S277^{6.48}W/L304^{7.42}A double mutation (b) using a GloSensor cAMP assay. c The expression levels of EP4, EP2 receptors and their mutants in cos7 cells measured by cell surface staining. Data are represented as mean ± SD of three independent experiments.

5) In line 192, the statement "switching the residues significantly impaired the binding of the ligands to the receptors" is not entirely accurate, as the signaling assays used in the study only evaluate receptor activation, not ligand binding. Similarly, the term "affinity" is used in several instances throughout the manuscript (e.g. lines 137, 149, 170, 177, 193), which specifically refers to binding, while only receptor activity was assessed in the experiments. Thus, it is important to be mindful of the terminology used when describing the result

We thank the reviewer for the suggestions. We used 'potency' or 'EC50' instead of 'affinity' in the manuscript.

6) In Fig. 5, the importance of three interactions for Gq selectivity is noted. However, the contribution of each interaction should be verified through mutagenesis experiments. Specifically, the authors have the capability to conduct NanoBiT G protein dissociation

assays, which would undoubtedly strengthen their argument by investigating the effects of mutations in the Gq protein.

We thank the reviewer for the suggestion. We performed the mutagenesis studies and the results were shown in the response to Reviewer 1 (Major 3).

7) Wu et al. also determined the structures of the FP receptor bound to $\text{PGF}_{2\alpha}$ and latanoprost free acid. If possible, it would be beneficial to compare and discuss whether there are any differences between their structures and the structures determined by the authors.

We have added the structure comparison in the discussion part of the revised manuscript as the reviewer suggested.

Minor points:

- Line 14: No. 5 affiliation is not included in the list of authors.

We have corrected this in the revised manuscript.

- Line 25: "female production" seems to be a typo for "female reproduction."

We have corrected this in the revised manuscript.

- Line 48: While it may be an editorial convenience, the final version should correctly subscript 2α in $\text{PGF}_{2\alpha}$.

We have corrected this in the revised manuscript as the reviewer suggested.

- Line 52: Carboprost tromethamine is described as a mixture of 15-methyl $\text{PGF}_{2\alpha}$ and tromethamine, but I think "salt" may be a more accurate term.

We changed the description to 'Carboprost tromethamine is a tromethamine salt form of 15-methyl $\text{PGF}_{2\alpha}$ '.

- Line 62: To my knowledge, the paper cited in reference 12 only shows the affinity of $\text{PGF}_{2\alpha}$, not carboprost specifically. Could you please indicate where carboprost is mentioned in the paper?

We thank the reviewer for catching this and apology for our careless mistake on reference citation. Indeed the reference only shows the affinity of $\text{PGF}_{2\alpha}$ and we did not find publications reporting the affinities of carboprost towards prostaglandin receptors. Our functional assay verified the similar functional profiles of carboprost and $\text{PGF}_{2\alpha}$ as shown in Response figure 9 (Supplementary Fig. 1 in revised manuscript). We have adjusted our description in the revised manuscript as follows.

"Both carboprost and $\text{PGF}_{2\alpha}$ only have 10-fold selectivity towards the FP receptor over the EP3 receptor in NanoBiT assay (Supplementary Fig. 1a, b), which is consistent with previous reports on $\text{PGF}_{2\alpha}$'s 10-fold selectivity towards the FP receptor²²."

Response Figure 9. The selectivity of PGF_{2α} and carboprost toward FP receptor and EP3 receptor. **a** The PGF_{2α} activation profiles of FP receptor and EP3 receptor revealed by NanoBiT assay. **b** The carboprost activation profiles of FP receptor and EP3 receptor revealed by NanoBiT assay.

- Line 80: The notations miniG_{s/q70iN} and miniG_{sqiN} are used multiple times in the manuscript. If they refer to the same thing, they should be unified.

We thank the reviewer for the suggestion. We have corrected this and used 'miniG_{s/q70iN}' in the revised manuscript.

- Lines 80 to 83: The authors should first mention the difference between the miniG_{s/q70} used in ref.14 and ref.15, which does not have the N-terminus replaced by Gi, and the miniG_{s/q70iN} used in their study. In this section, it should also be noted that only the 7 residues in the α5 helix were replaced with those from Gq. Furthermore, the Materials and Methods section should clearly state the precise sequence of the substitution.

We have added the description on the difference between miniG_{s/q70iN} used in this study and the miniG_{s/q70} used in ref.14 and ref.15 in the revised manuscript. We also mentioned that only the 7 residues in the α5 helix were replaced with those from Gq, and added the sequence of the substitution in the Materials and Methods in the revised manuscript.

- Lines 146 to 147: In reference 12, Table 2 shows the K_i values for PGF_{2α} towards EP3 and FP as 38 nM and 3.2 nM, respectively, and the K_i values for PGE₂ towards EP3 and FP as 0.33 nM and 119 nM, respectively. Based on these values, PGF_{2α} and PGE₂ appear to have selectivity values of 12-fold and 360-fold, respectively. However, there is a discrepancy between the selectivity values mentioned by the authors and my calculated results. Can you please clarify which values were used for the authors' calculations?

We thank the reviewer for catching this mistake. We have corrected this in the revised manuscript.

- Line 209: As noted by the authors, "latanoprost" specifically refers to the isopropyl ester form, which is a prodrug. Therefore, it is recommended to use a different term such as "latanoprost-FA" to refer to the free acid form.

We agree with the reviewer and use the term 'latanoprost-FA' in the revised manuscript.

- Line 378: The correct full name of Ric-8A is resistance to inhibitors of cholinesterase-8A.
We have corrected this in the revised manuscript.

- Line 592: The authors with the same initials X.Z. should be represented in a way that allows for a clear distinction between the two individuals.

We have corrected this in the revised manuscript.

- Line 665: The manuscript mentions "3 independent experiments", but in Supplementary Fig. 5c, there appear to be around 8 dots.

We have corrected the figure legend in the revised manuscript to 'Data are given as mean \pm SD from 3 (b) or 8 (c) independent experiments.'

- Line 704: "hydrophic" seems to be a typo for "hydrophilic."

We have corrected this in the revised manuscript.

- Fig. 1a: The arrow between "uterine" and "PPH" is unclear in its meaning. Could you please provide an explanation?

We thank the reviewer for the suggestion. We initially wanted to show that both FP receptor and EP3 receptor are expressed on uterine. But we agree this arrow is confusing at its position and we delete it in the revised manuscript.

- Fig. 1b: The label/number for 'b' is missing in the figure.

We have added the number 'b' in the Fig. 1 in the revised manuscript.

- Fig. 2c–h, Supplementary Fig. 2c: It is confusing because it appears that activation is suppressed as the ligand concentration increases. Consider changing the Y-axis to G protein dissociation or setting the activation state value to 1.

We have changed the Y-axis to G protein dissociation as the reviewer suggested.

- Fig. 4a: The double bond between positions 5 and 6 in latanoprost-FA is in the cis configuration.

We have corrected this in the revised manuscript.

- Supplementary Fig. 3: The residues before and after 5.41/5.43 are reversed in all receptors.

We thank the reviewer for catching this mistake. We have corrected it in the revised manuscript.

- Supplementary Fig. 4a: The stereochemistry at position 15 of PGE₂ and PGF_{2 α} should be indicated for clarity.

We have indicated the stereochemistry at position 15 of PGE₂ and PGF_{2 α} in the supplementary figure 7a of the revised manuscript.

- Supplementary Fig. 7d, e: To facilitate comparison of carboprost-bound and latanoprost-bound forms, they should be shown as oriented as possible. For example, Y92^{2.65} and W262^{6.48} are visible in d but not in e.

We have adjusted the figures in the revised manuscript (new Supplementary Figure 9d, e) as the reviewer suggested.

- Supplementary Tables. 1, 2: pEC50 is defined as the negative common logarithm of EC50, so the signs of the values in Supplementary Table 1 are expected to be opposite to what is shown. Similarly, the vertical axis of Fig. 4e and Supplementary Fig. 5b appears to be inverted.

We thank the reviewer for the suggestion and have corrected this in the revised manuscript.

-A schematic representation of carboprost recognition, similar to Fig. 4a, should also be included for clarity.

We agree with the reviewer and have added a schematic representation of carboprost recognition in the supplementary figure 5a of revised manuscript (Response figure 10).

Response figure 10. The schematic representation of carboprost recognition.

Reviewer #3 (Remarks to the Author):

Postpartum hemorrhage (PPH) accounts for approximately 18% of all deaths of pregnant women globally and represents a leading cause of maternal morbidity and mortality. Carboprost tromethamine is the most effective treatment of atonic PPH by activating the prostaglandin F2-alpha receptor (FP receptor), however, it has side effects such as fever and hypertension due to its activation of the closely related EP3 receptor. The off-target side effects limit its clinical application, especially to patients with cardiovascular diseases. In this manuscript, Lv et al. reported the high-resolution cryo-EM structures of the human

prostaglandin F2-alpha receptor (FP receptor) bound with carboprost or a more selective drug latanoprost. The structural information reveals the molecular mechanism underlying the FP receptor selectivity of these clinical drugs and provides guidance for the development of better PPH drugs. The work is of general interest to the field.

We thank the reviewer for the positive comments on this work.

I have the following minor suggestions:

1. The label for panel b is missing in figure 1.

We have added the label 'b' in the Figure 1.

2. The labeling in Figure 5a is too small. The authors should adjust the size of the labelling.

We have adjusted the character size in the revised manuscript as the reviewer suggested.

3. The cryo-EM data processing flow-chart (supplementary figure 7) lacks the essential details. For example, how many particles were used to calculate the final maps for each structure? Also in panel d and e, the readers may understand these are density maps for TM1-7, but it would be more clear if these helices are labelled. Please show densities of the residues in orthosteric site, H8 helix, and extra- or intra-cellular loops.

We have added the information in the revised manuscript.

4. The authors may consider using the template provided by Nature to prepare the cryo-EM data collection and processing table. (<https://www.nature.com/nature/for-authors/formatting-guide>)

We thank the reviewer for the suggestion. We have prepared the cryo-EM data processing table using the template provided by Nature in the revised manuscript.

Reviewer #1 (Remarks to the Author):

The authors addressed my concerns. I recommend its publication.

Reviewer #2 (Remarks to the Author):

The manuscript by Liu and co-workers has shown significant improvement. The authors have addressed nearly all of the relevant issues raised in response to my previous questions. However, I still have one point of concern.

The authors assert that 11-deoxy PGF_{2α} was not available from the market. However, I have found that it is indeed available in my country through the Cayman Chemical Company. I am curious as to why the authors were unable to obtain it from this source. The use of 11-deoxy PGF_{2α} for investigating FP receptor selectivity is a critical aspect of the paper's significance. Nonetheless, if there are legitimate reasons for its unavailability, I will acknowledge this limitation.

Additionally, I would like to raise the following minor concerns:

- Line 49: The number "2" is missing after prostaglandin D.
- Line 61: The abbreviation "FP" is already mentioned in line 48.
- The size of Supplementary Fig. 10b,c appears to be too small.

Reviewer #3 (Remarks to the Author):

The authors have addressed my concerns.

We appreciate the suggestions from the reviewers and have addressed all the comments. The point-by-point responses to the concerns/suggestions are provided below in blue font.

Reviewer #1 (Remarks to the Author):

The authors addressed my concerns. I recommend its publication.

We thank the reviewer for his/her suggestions to help us improve the manuscript.

Reviewer #2 (Remarks to the Author):

The manuscript by Liu and co-workers has shown significant improvement. The authors have addressed nearly all of the relevant issues raised in response to my previous questions. However, I still have one point of concern.

We thank the reviewer for his/her suggestions to help us improve the manuscript.

The authors assert that 11-deoxy PGF_{2α} was not available from the market. However, I have found that it is indeed available in my country through the Cayman Chemical Company. I am curious as to why the authors were unable to obtain it from this source. The use of 11-deoxy PGF_{2α} for investigating FP receptor selectivity is a critical aspect of the paper's significance. Nonetheless, if there are legitimate reasons for its unavailability, I will acknowledge this limitation.

We agree with the reviewer that 11-deoxyl PGF_{2α} would indeed serve as an excellent tool compound for exploring FP receptor selectivity. We consulted different companies including Cayman Chemical Company, but without success. Specifically, the agent representing Cayman Chemical informed us that this company no longer has cooperate offices in China. As a result, there are only two small-molecule products left in stock from this company, while the rest products are unavailable.

Additionally, I would like to raise the following minor concerns:

- Line 49: The number "2" is missing after prostaglandin D.

We thank the reviewer for the careful reading. We have corrected it in the revised manuscript.

- Line 61: The abbreviation "FP" is already mentioned in line 48.

We have corrected it in the revised manuscript.

- The size of Supplementary Fig. 10b, c appears to be too small.

We have adjusted the size of Supplementary Fig. 10b, c in the revised manuscript.

Reviewer #3 (Remarks to the Author):

The authors have addressed my concerns.

We thank the reviewer for his/her suggestions to help us improve our manuscript.